# LEARNING GENERALIZABLE AND WELL-SHAPED REWARD FUNCTIONS FROM TOO FEW DEMONSTRATIONS

## ABSTRACT

Inverse reinforcement learning (IRL) is an important problem that aims to learn a reward function and policy directly from demonstrations, which can often be easier to provide than a well-shaped reward function. However, many real-world tasks include natural variations (i.e., a cleaning robot in a house with different furniture configurations), making it costly to provide demonstrations of every possible scenario. We tackle the problem of few-shot IRL with multi-task data where the goal is for an agent to learn from a few demonstrations, not sufficient to fully specify the task, by utilizing an offline multi-task demonstration dataset. Prior work utilizes meta-learning or imitation learning which additionally requires reward labels, a multi-task training environment, or cannot improve with online interactions. We propose Multitask Discriminator Proximity-guided IRL (MPIRL), an IRL method that learns a generalizable and well-shaped reward function by learning a multi-task generative adversarial discriminator with an auxiliary proximity-to-expert reward. We demonstrate the effectiveness of our method on multiple navigation and manipulation tasks.

## 1 INTRODUCTION

Reinforcement Learning (RL) has shown impressive results in learning sequential decision-making tasks from scratch by optimizing a pre-defined reward function (Sutton & Barto, 2018). While this is a general framework with powerful algorithms, the need for a reward function for each task, which often needs to be hand-specified and well-shaped (Amodei et al., 2016; Gupta et al., 2022; Rengarajan et al., 2022), requires significant human effort. Inverse Reinforcement Learning (IRL) (Ng & Russell, 2000) offers an alternative by learning directly from expert demonstrations, inferring the reward function instead of requiring it to be manually defined. Consider a household robot capable of performing basic cleaning tasks such as sweeping and cleaning countertops. When learning a new task, like vacuuming, the robot should be able to infer the objective of the task from a couple of demonstrations without needing a fully defined reward function or being shown how to operate the vacuum in every room of

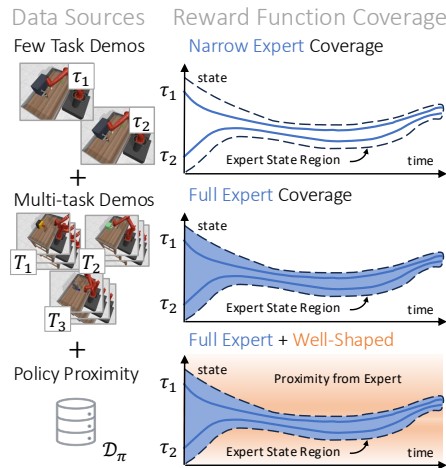

Figure 1: We learn a generalizeable and well-shaped reward by making use of multi-task demonstrations and policy proximity.

the house. From its experience sweeping, the robot can infer that it should navigate around different furniture configurations while vacuuming. Similarly, we aim to tackle IRL given expert demonstrations that *are too few to fully specify the task* in every environment setting by utilizing the agent's multi-task knowledge. This problem setting greatly reduces the burden of task specification in tasks with natural variations while using existing powerful RL algorithms.

Prior work in utilizing multi-task information to do few-shot IRL use meta-learning (Xu et al., 2019; Yu et al., 2019; Seyed Ghasemipour et al., 2019), which requires training over multi-task environments and/or access to the multi-task reward functions, or imitation learning without learning

a reward function (Dance et al., 2021; Hakhamaneshi et al., 2021; Finn et al., 2017), which limits the agent's ability to improve through additional trials. On the other hand, many few-shot imitation learning works learn a more well-shaped reward function with proximity-based rewards (Dadashi et al., 2021; Haldar et al., 2022; Chiang et al., 2024) but do not address the challenge of generalization across task variations. Instead, we propose a novel IRL problem setting where an agent has access to a few expert demonstrations of a task with variations, a large multi-task dataset of other demonstrations, and access to a training environment for the current task. Most closely related to our work, Chen et al. (2021) learns a generalizable multi-task video success discriminator from a few robot demonstrations and a dataset of human demonstrations but does not learn an RL policy.

We propose Multitask Discriminator Proximity-guided IRL (MPIRL), a novel few-shot IRL method that addresses the challenge of learning a reward function and RL policy from too few demonstrations, which cannot fully specify the task in an environment with variations, by making use of a multi-task dataset of expert trajectories. A generalizable and well-shaped reward function must infer two things from the demonstrations: 1) What does expert behavior look like in different task variations? and 2) How to shape the reward in non-expert states to guide the policy towards expert behavior?. We propose a two-part reward function consisting of 1) a generalizable multi-task discriminator that uses the multi-task data to infer expert behavior across task variations and 2) a proximity reward function that predicts how many steps the agent is away from the expert state distribution, helping guide the agent toward expert states. While the multi-task discriminator reward alone could theoretically provide this guidance, we found that the proximity reward conferred significant improvements in sample efficiency and final performance by offering a smooth and dense reward that encourages the agent to stay near the expert trajectory distribution (see Figure 1).

We propose the problem setting of few-shot IRL with multi-task demonstrations and identify the challenge of under-specification in realistic tasks with natural variations. Our main contribution is a MPIRL, a novel method that enables IRL with too few demonstrations that do not fully specify the task by leveraging a multi-task demonstration dataset and learning a generalizable and well-shaped reward function. Our experimental results on maze navigation, block stacking, and manipulation tasks in FactorWorld (Xie et al., 2024), demonstrate the effectiveness of our method, achieving an average 33% success rate improvement over the next best-performing method.

## 2 RELATED WORK

### 2.1 FEW-SHOT IMITATION LEARNING

Imitation learning aims to replicate expert behavior by learning directly from expert demonstrations. While closely related to IRL, in this paper, we distinguish pure imitation learning by methods that learn a policy directly without inferring or optimizing a reward function. Early approaches for addressing few-shot imitation learning focus on Behavior Cloning (BC) (Finn et al., 2017; Duan et al., 2017; Yu et al., 2018), which compares predicted actions with those from demonstrations using loss functions like mean squared error or cross-entropy loss. Dance et al. (2021) learn a demonstration-conditioned policy but requires access to multi-task training environments and corresponding reward functions. Hakhamaneshi et al. (2021) extract skills and an inverse skill dynamics model from a large offline dataset to facilitate few-shot imitation learning. Other works explore offline imitation learning utilizing a large offline dataset similar to our work (Luo et al., 2023; Xu et al., 2022; Chang et al., 2021), but these works do not explicitly address the challenge of few-shot imitation. However, overall, imitation learning methods suffer from compounding errors over time and cannot improve through additional online interactions without learning a reward function. In response to this, Reddy et al. (2020) propose a simple, sparse reward label to allow for policy optimization through RL. Meanwhile, Chae et al. (2022) addresses environment dynamic variations by imitating multiple experts in different environment dynamics.

### 2.2 FEW-SHOT INVERSE REINFORCEMENT LEARNING

The most common approach to few-shot IRL is through meta-learning, which meta-trains a context-conditioned reward function (Yu et al., 2019; Seyed Ghasemipour et al., 2019) or learns a good initialization for reward function training (Xu et al., 2019), using traditional IRL algorithms (Ziebart et al., 2008; Fu et al., 2018). These methods, however, often require access to multi-task environ-

ments or transition functions to train in, which may not be feasible if task environments differ. In contrast, our approach only necessitates access to the environment of the target task. It aims to leverage the variations in the multi-task demonstration dataset to learn a generalizable reward function. This results in a more sample-efficient and practical solution in real-world settings where data collection and computational resources are constrained.

Chen et al. (2021) propose DVD, a multi-task video discriminator trained on a large, diverse human video dataset capable of generalizing across task variations from a few robot demos, but does not employ RL to learn a task policy. Xie et al. (2018) develop a success classifier for goal-conditioned tasks from a few examples, but they do not learn a full reward function. Our work can be viewed as an extension of these ideas to the multi-task setting and learning a reward function suitable for online RL. Other works have explored demonstration-efficient IRL in multi-task (Gleave & Habryka, 2018) and multi-agent (Filos et al., 2021) settings.

### 2.3 Proximity-based Rewards

Popular and practical IRL methods including Ho & Ermon (2016) and Fu et al. (2018) learn a reward function by discriminating between agent and expert behaviors, typically through binary classification. However, the rewards learned this way may not provide sufficiently rich signals to guide agents in non-expert states, especially in the few-shot setting. To address this limitation, recent works have proposed different forms of reward shaping that estimate some form of proximity to the expert. This includes a progress estimator for goal-conditioned tasks (Lee et al., 2021), Euclidean distance between the agent's and expert's state-action pairs (Hakhamaneshi et al., 2021), and geometric distance functions that measure the difference between the agent and the expert distribution (Dadashi et al., 2021; Haldar et al., 2022). Chiang et al. (2024) learns a *transition discriminator* that approximates whether one state can reach another within a single step in order to reward the agent based on the likelihood that it is one step away from an expert trajectory. While these methods provide useful guidance in non-expert states, they do not account for generalization across task variations with too few demonstrations, limiting the agent's ability to recover and return to the expert distribution when task variability increases.

## 3 Problem

Inverse RL addresses the problem of learning sequential decision-making tasks from demonstration. We consider these tasks to be Markov decision problems (MDPs) defined by the tuple $(\mathcal{S}, \mathcal{A}, \mathcal{T}, \rho, \mathcal{R})$: state space $\mathcal{S}$, action space $\mathcal{A}$, transition probabilities $\mathcal{T}$, initial state distribution $\rho_i$, and underlying reward function $\mathcal{R}$. We assume $\mathcal{R}$ is not available and instead must be inferred from a set of demonstrations $\mathcal{D}$ from an expert policy $\pi^*(a|s)$. The goal is to learn a reward function $\tilde{\mathcal{R}} : \mathcal{S} \times \mathcal{A} \to \mathbb{R}$ that approximates the true $\mathcal{R}$ and policy $\pi(a|s)$ that approximates $\pi^*(a|s)$.

**Few-Shot IRL with Multi-task Dataset**: In our problem setting, we consider the specific case of few-shot IRL, that is when there are too few demonstrations available to fully specify the desired behavior in all instances of the task. This can easily happen when there is task variation occurring from the initial state distribution $\rho_i$, for example, variations in the initial state of the agent or objects it interacts with in the environment. Therefore, doing naive imitation learning or IRL on these demonstrations will fail in task instances outside of those seen in $\mathcal{D}$. Our goal is to do few-shot IRL given a large offline dataset of multi-task demonstrations for $T$ tasks $\mathcal{D}_{multitask} = \{\mathcal{D}_1, \mathcal{D}_2, \cdots, \mathcal{D}_T\}$ of the same agent doing different tasks in the environment with similar task variations. Formally, each task $i$ has a distinct underlying reward function $\mathcal{R}_i$ and initial state distribution $\rho_i$, which can include different environment layouts and object positions, but shares the same $(\mathcal{S}, \mathcal{A}, \mathcal{T})$ as the other tasks and the target task. Practically, $\mathcal{D}_{multitask}$ can be gathered from an agent's prior experience in a multi-task or continual learning setup where rewards are available using a well-trained RL policy.

## 4 Our Method: Multitask Discriminator Proximity-guided IRL

The few expert demonstrations for the target task $\mathcal{D}$ are insufficient to infer the desired behavior in every task instance. For example, we would like our household robot to learn to vacuum the entire house from a couple of demonstrations of vacuuming the living room. The robot should be able to

infer how it should vacuum all rooms in the house based on its experience sweeping and cleaning some rooms. Similarly, we utilize the large offline multi-task dataset $\mathcal{D}_{multitask}$ to infer what the target task reward function might look like beyond the narrow support of the few demonstrations. Our main insight comes from decomposing the reward function into two components that are easier to learn on their own: 1) What is the desired behavior in unseen task instances (i.e., what the expert trajectory distribution is)? and 2) What should the reward function look like in non-expert states to guide the policy towards expert behavior? Our final reward function is the sum of the two components, a multi-task discriminator-based reward (Section 4.1) and a proximity reward (Section 4.2), that utilizes $\mathcal{D}_{multitask}$ and is trained with a task policy $\pi$ to successfully learn a generalizable and well-shaped task reward from only a few demonstrations. Figure 2 and Algorithm 1 summarize our method.

## 4.1 MULTI-TASK DISCRIMINATOR

Our multi-task discriminator builds on a generative adversarial backbone to align the policy distribution to the expert's, following GAIL (Ho & Ermon, 2016), and learns a multi-task discriminator like DVD (Chen et al., 2021). Specifically, we train a multi-task discriminator that takes as input a task demonstration, the current state and action, and predicts whether the state-action tuple belongs to the expert trajectory distribution for the demonstrated task, using binary classification loss, as described in Equation 1. We train the multi-task discriminator reward $D(\tau, s, a)$ adversarially with a policy by sampling target task demonstrations and corresponding state-action tuples as positive training samples and policy-generated state-action tuples as negative samples.

In addition, we extend the training across all tasks in $\mathcal{D} \cup \mathcal{D}_{multitask}$, where demonstration trajectories and state-action tuples from the same task are treated as positives, while state-action tuples from different tasks or from the policy are treated as negatives. Policy behaviors are always considered negatives for any task following the GAIL objective. By incorporating $\mathcal{D}_{multitask}$, the discriminator is able to learn a reward function for the target task that generalizes across task variations by observing similar task variations in other tasks. However, a well-trained discriminator tends to assign uniformly low rewards for all policy-generated samples outside of expert behavior, which fails to provide an adequate learning signal for an imperfect policy. For simplicity, we will use the notation $D(s, a)$ from now on to represent the target task discriminator, where we sample a target demonstration from $\mathcal{D}$ uniformly as the input demonstration.

$$L_D = \underbrace{\mathbb{E}_{\tau \sim \mathcal{D}, (s,a) \sim \pi}[\log(D(\tau, s, a))] + \mathbb{E}_{\tau, (s,a) \sim \mathcal{D}}[\log(1 - D(\tau, s, a)]}_{\text{Task-specific Adversarial Training}}$$
$$+ \underbrace{\mathbb{E}_{\tau \sim \mathcal{D}_i, (s,a) \sim \mathcal{D}_{j \neq i}, \pi}[\log(D(\tau, s, a))] + \mathbb{E}_{\tau, (s,a) \sim \mathcal{D}_i}[\log(1 - D(\tau, s, a)]}_{\text{Multi-task Training}} \quad (1)$$

## 4.2 PROXIMITY REWARD

To provide well-shaped rewards in non-expert states that guide the policy back towards the expert state distribution, we introduce a proximity reward $P(s)$, which penalizes states based on the temporal distance to the expert state distribution. Specifically, the proximity of a state $s$ is defined as the number of steps it takes the policy to get from $s$ to an expert state, defined by it being in $\mathcal{D}$ or the corresponding state-action tuple from the policy being classified as expert behavior by the target task discriminator $D(s, a)$. We define the proximity reward $P(s)$ to be inversely proportional to this temporal distance, scaled by a discount factor $-\gamma$, which should be set proportional to the episode horizon of the task. The reward $P(s)$ achieves a maximum value of 0 at expert states and decreases to a minimum value of $-1$ for unreachable states. The target proximity reward and its corresponding mean squared loss are formalized in Equation 2. Intuitively, $P(s)$ penalizes the policy for reaching states where it is difficult to return to the expert distribution, therefore guiding the policy in non-expert states.

$$L_P = \mathbb{E}_{s \sim \pi \cup \mathcal{D}}[(P(s) - (-\gamma \cdot \# \text{ steps to expert state}))^2] \quad (2)$$

Figure 2: Our method learns a generalizable and well-shaped reward function from a few target task demonstrations by learning a reward function composed of a multi-task discriminator and a proximity reward. We combine these rewards into $\tilde{R}$ which we use to train a policy with RL.

However, the exact number of steps to an expert state is challenging to determine and can change as the policy explores more of its environment. To get the most accurate labels for the proximity function, we propose to generate pseudo-labels $prox(s)$ by continually re-labeling the training dataset with the updated $P(s)$. Specifically, we calculate the proximity at time $t$ as $prox(s_t) = P(s_{t+1}) - \gamma$, since state $s_t$ is one policy step further from the expert than state $s_{t+1}$. If $s_t$ itself is an expert state, determined by $D(s_t, a_t) > c_{thresh}$ for some fixed threshold value, we label it with $prox(s_t) = 0$. Unfortunately, directly relabelling each state recursively like this results in degenerate training because the pseudo-labels become too similar to $P(s)$, causing $P(s)$ to predict itself. Instead, we randomly sample a batch of trajectories from the proximity dataset, consisting of the multi-task demonstrations and policy samples, and sample a state-action tuple $(s_t, a_t)$ from each trajectory. We then predict the label on these samples and perform *backwards re-labeling* for earlier states in the trajectory using $prox(s_{t-k}) = P(s_t) - \gamma k$. This random sampling and backwards re-labeling strategy balances the accuracy and stability of the pseudo-labels. Equation 3 details the full pseudo-label updates at training step $i$.

$$prox_i(s_t) = \begin{cases} 0 & \text{if } s_t \in \mathcal{D} \text{ or if } D(s_t, a_t) \\ P(s_t) & \text{otherwise} \end{cases}$$

$$prox_i(s_k) = P(s_t) - \gamma k \text{ for } k = 0, 1, \cdots, t-1$$

(3)

### 4.3 MULTITASK DISCRIMINATOR PROXIMITY-GUIDED IRL

We combine the multi-task discriminator and proximity function into our reward function Equation 4, applying a scaling factor of $\lambda_{prox}$.

$$\tilde{R}(s, a) = D(s, a) + \lambda_{prox} P(s) \quad (4)$$

We use $\tilde{R}(s, a)$ to label the trajectories collected by the policy so we can optimize the policy with any RL algorithm. During online training, we iteratively train the policy $\pi$, discriminator reward $D$, and proximity reward $P$, which is detailed in Algorithm 1. The multi-task discriminator $D$ is trained adversarially with the policy $\pi$ and uses the multi-task demonstrations to learn to classify expert behavior across task variations. The proximity reward $P$ is updated through

---

**Algorithm 1** MPIRL

**Input:** Task Demos $\mathcal{D}$, Multi-task Demos $\mathcal{D}_{multitask}$
Initialize $\pi$, Discriminator $D$, Proximity $P$, Replay buffer $\mathcal{D}_\pi$, Proximity dataset $\mathcal{D}_{prox}$
**for** each epoch **do**
    Gather a batch of data $(s_t, a_t, s_{t+1})_{t=0}^N$ by rolling out $\pi$ in the task environment
    Append to $\mathcal{D}_\pi$ and $\mathcal{D}_{prox}$
    Update $\pi$ with RL, reward labels from Eqn. 4
    Update $D$ with Eqn. 1
    Update $P$ with Eqn. 2
    Relabel $\mathcal{D}_{prox}$ using Eqn. 3
**end for**
**Output:** Trained policies $\{\pi_i\}_{i=1}^N$

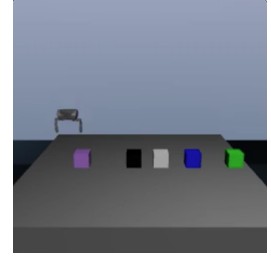 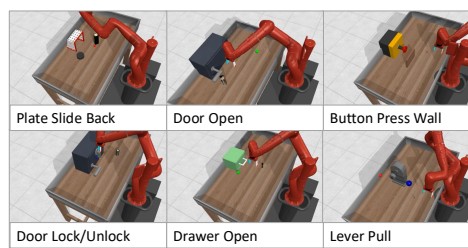

| (a) Maze2D | (b) Block Stacking | (c) FactorWorld |

Figure 3: **Environments & Tasks**: (a) Maze2D. The randomly initialized agent must reach different goals. (b) Block Stacking. The agent must pick up one color block and stack it on top of another color block. (c) FactorWorld. Multiple table-top manipulation tasks from Meta-World.

its re-labeling process and as $\pi$ changes to more accurately estimate the policy's temporal distance to the expert. The policy is trained with RL to optimize this combined reward, which rewards it for mimicking the expert distribution and penalizes it for straying too far. In summary, MPIRL enables IRL with too few demonstrations by utilizing a multi-task demonstration dataset to infer an accurate and well-shaped reward function.

## 5 EXPERIMENTS

We evaluate our methods on several IRL tasks in different environments (Figure 3) and compare with multiple baseline methods described below. For further details, see Appendix Section B for environment and Appendix Section C for baseline implementation.

### 5.1 ENVIRONMENTS

- **Maze2D** We first examine our method in *Maze2d* from the D4RL benchmark (Fu et al., 2020). The goal for each task is to navigate to a certain colored ball whose positions are fixed as shown in Figure 3a. The starting position of the agent is randomly sampled, creating task variation. We use two demonstrations of the target task and a multi-task dataset consisting of 200 demonstrations for each of the three other Maze tasks.

- **Block Stacking** On the *Block Stacking* task (Pertsch et al., 2021), there are five colored blocks whose positions are randomly initialized, creating task variation. In each task, the agent aims to pick up a block with color X and place it on a block with color Y. We use 25 target task demonstrations and collect 200 demonstrations for three other tasks.

- **FactorWorld** from Xie et al. (2024) is a multi-task benchmark of manipulation tasks with variations in object position, table position, distractor objects & positions, and arm position. We evaluate on 7 different target tasks with 2 to 25 demonstrations depending on the task. The multi-task demonstration dataset consists of 10 tasks with 200 demonstrations each.

### 5.2 BASELINES

To the best of our knowledge, there is no prior work that tackles our exact problem setting: few-shot IRL with a multi-task demonstration dataset. So we compare with SOTA methods in similar problem settings and provide them with additional assumptions where possible for a fairer comparison. All online methods use PPO as the RL algorithm except SQIL which uses the off-policy algorithm SAC.

- **BC** behavior clones the few task demonstrations and does not utilize multi-task demonstrations or environment interactions.

- **GAIL** (Ho & Ermon, 2016) learns a policy and reward function adversarially. In our GAIL experiments, we use the multi-task demonstrations as additional non-expert samples.

- **DVD** (Chen et al., 2021) learns a multi-task discriminator reward function using all demonstrations. We evaluate by training a policy using this reward for online RL.

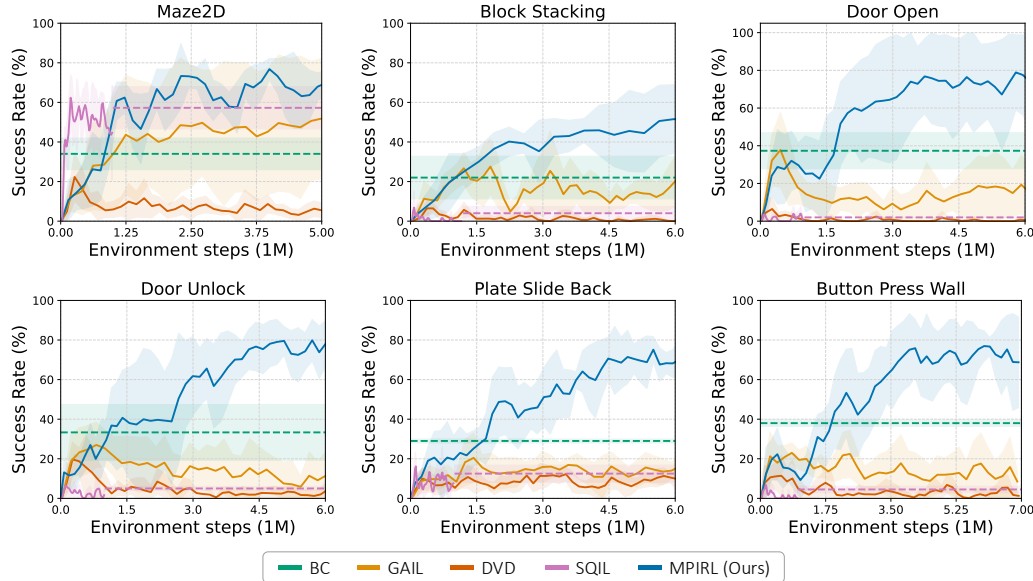

Figure 4: MPIRL (blue) achieves better performance compared to other imitation learning and IRL methods. We plot the average and standard deviation (in shaded regions) over 4 seeds per method and roll out 10 episodes per evaluation. For BC and SQIL, the dashed lines represent the performance at convergence. See Appendix Figure 7 for additional tasks.

- **SQIL** (Reddy et al., 2020) is an imitation learning algorithm using RL by labeling with sparse rewards. Similar to our GAIL implementation, we use the multi-task demonstrations and label it with with 0 reward. Note: SQIL converges more quickly and takes longer to run than other methods due to using SAC so we only train until convergence.

## 6 RESULTS

We answer the following questions in our experiments: (1) How effective is MPIRL compared to other methods that learn from demonstrations? (2) How does MPIRL's performance vary with the number of target demonstrations and quality of the multi-task dataset? (3) Ablations on MPIRL.

### 6.1 COMPARISON

To evaluate the effectiveness of our method, we compare against multiple imitation learning and IRL methods in nine tasks over three different simulated environments: Maze2D, Block Stacking, and seven tasks in FactorWorld. We demonstrate in Figure 4 (additional tasks in Appendix Figure 7) that across all tasks, MPIRLconsistently outperforms other methods and achieves an average success rate of 33% over the next best performing method. Moreover, compared to other methods that use online RL, our method is able to more consistently improve in success rate over additional trials, demonstrating our learned reward is better suited for RL.

In Maze2D, SQIL performs comparably with MPIRL, likely due to the large multi-task demonstration dataset providing sufficient coverage of the maze environment to learn a good policy from a few demonstrations. However, GAIL, which uses the multi-task demonstrations similarly but learns an adversarial reward function/discriminator, is the worst-performing method, illustrating the potential instability of learning a reward function, especially through adversarial training. While our method also utilized the GAIL objective in training the discriminator part of the reward function, the addition of the multi-task discriminator and the proximity reward make the reward function much more stable for RL as we will discuss further in Section 6.3.

Block Stacking is a challenging task for imitation learning, requiring 25 demonstrations for our method to reach a 50% success rate, still double that of the next best baseline. This task is likely more challenging because it is less forgiving: dropping a block at the wrong time quickly takes the policy out of distribution and is almost always unrecoverable. We hypothesize that our proximity

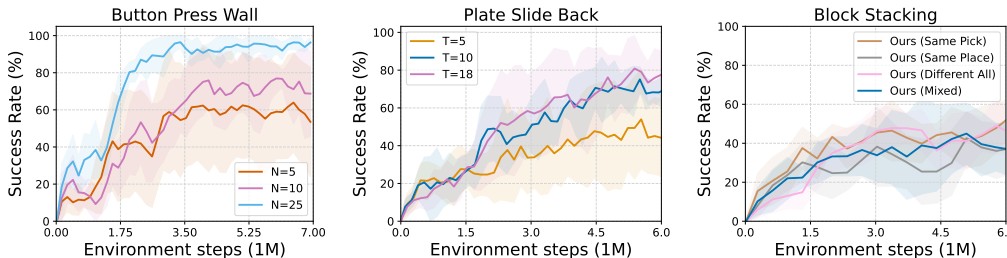

(a) Number of Target Task Demos  (b) Number of Tasks in $\mathcal{D}_{multitask}$  (c) Types of Tasks in $\mathcal{D}_{multitask}$

Figure 5: We study our method by varying (a) the number of target task demonstrations $N$ we provide, (b) the number of tasks $T$ in the multi-task dataset, and (c) the types of tasks in the multi-task dataset where SAME PICK and SAME PLACE share more similarities with the target task.

reward partially addresses this by penalizing those states more than other less harmful non-expert behaviors, since the expert distribution is often completely unreachable after dropping the block.

In FactorWorld, each task differs semantically (i.e., opening a door vs. pressing a button) and the environment setup differs depending on which objects must be present on the tabletop. SQIL and DVD both perform poorly. Since each task's environment is different, using the multi-task data directly for policy training may not be as useful. DVD has a fixed reward function that could be exploited; the addition of an online adversarial objective (see Section 6.3) improves it significantly but still underperforms our full method. Meanwhile, BC is a surprisingly strong baseline even in the too-few-demonstrations regime, attaining up to 35% success with just 2 demonstrations in Maze2D with a uniformly randomized start position. This additionally highlights the challenge of IRL from a few demonstrations: it becomes more difficult to infer a good reward function for RL rather than learn a reasonable BC policy that does not generalize over all task variations. This is why utilizing the multi-task demonstration dataset to generalize across task variations and adding the proximity reward for reward shaping is crucial to MPIRL's ability to infer a good reward function.

## 6.2 ANALYSIS

To understand how our method operates under different data conditions, we look at how our method performs by varying the number of target task demonstrations we have access to, the number of tasks in the multi-task demonstration dataset, and how similar those tasks are to the target task. As we see in Figure 5a, predictably MPIRL's performance on the FactorWorld Button Press Wall task increases as we provide more task demonstrations, with the performance jumping 20% as we increase from 5 to 10 demonstrations and saturating at around 25 demonstrations, which shows how MPIRL scales well with a modest number of additional demonstrations. In Figure 5b, we vary the number of tasks $T$ in the multi-task demonstration dataset, increasing the size and diversity of the dataset. We see that the performance increases with $T$ up until $T = 10$. Increasing to $T = 18$ did not change the performance significantly. MPIRL scales with the number of tasks in the multi-task dataset only to a point where additional tasks do not provide any more information helpful to the target task. Finally, we vary how similar the tasks in the multitask demonstrations dataset are in Block Stacking, as detailed below.

- SAME-PICK: All tasks require picking up the same colored block as the target task.

- SAME-PLACE: All tasks require placing on the same colored block as the target task.

- DIFFERENT ALL: No task shares the same colored block to be picked up or placed on as the target task.

- MIXED: The default dataset with some shared pick and place blocks.

In Figure 5c, we see that there is no significant difference in performance. Since we use these task demonstrations to learn a multi-task discriminator and not the target task reward or policy directly, our method does not require that these demonstrations share goals or behaviors with the target task, only that they exhibit similar types of task variation.

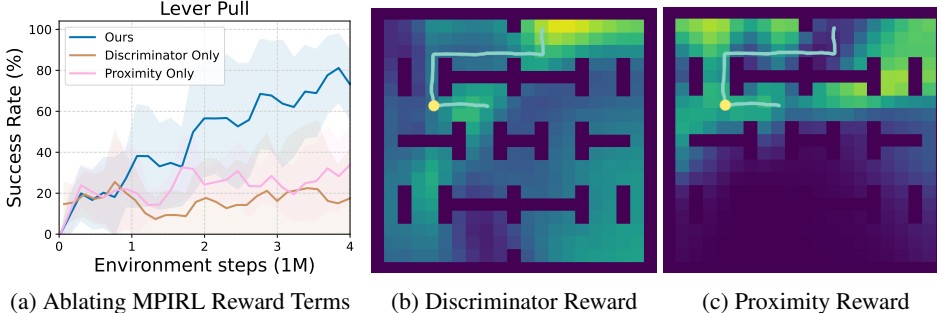

(a) Ablating MPIRL Reward Terms     (b) Discriminator Reward     (c) Proximity Reward

Figure 6: (a) We ablate the two reward terms in MPIRL to see that both components are necessary. In (b) and (c), we visualize the two components of the reward function during training in Maze2D. Lighter colors represent higher values. The task goal and demonstrations are also illustrated.

## 6.3 ABLATIONS

We ablate the two parts of our reward function $\tilde{R}(s, a) = D(s, a) + \lambda_{prox} P(s)$ by training a policy with DISCRIMINATOR ONLY reward or PROXIMITY ONLY reward. We see in Figure 6a that while each part of the reward function provides benefits on its own, both are necessary for the best performance for MPIRL. Therefore, the two parts of the reward function must provide some complementary information for the reward function like we hypothesized. The multi-task discriminator helps learn a generalizable reward that can recognize expert task behavior in different task variations while the proximity reward provides a well-shaped reward in non-expert states that guides the policy towards expert behavior.

To study the difference between these reward functions qualitatively, we visualize the two parts of the reward, multi-task discriminator reward (Figure 6b) and proximity reward (Figure 6c), in the maze environment. The discriminator reward is dense over the entire maze since it is trained on the multi-task demonstrations to generalize across different task variations, which in this task is the initial position of the agent. Meanwhile the proximity reward is low in the bottom half of the maze, likely due to the policy not finding a way to the goal from that half of the maze yet. However, it provides a well-shaped reward in the top half that steers the policy away from corners where it can get stuck and the bottom of the maze. Although neither reward is perfect, due to the few target task demonstrations and this being a snapshot taken during training, this demonstrates that both parts of the reward contribute differently to MPIRL.

## 7 LIMITATIONS

While MPIRL is capable of learning new tasks from scratch using only a few demonstrations, there are still challenges in applying our method to real-world scenarios. MPIRL requires a structured multi-task demonstration dataset in order to infer task variations. To relax this assumption and make use of unstructured data, one solution is to replace the requirement for task labels by using latent intention modeling, as proposed by Hausman et al. (2017) or making use of pre-trained large language or vision models like Sontakke et al. (2023). Additionally, we assume all our demonstrations come from the same domain and agent. Recent advancements in cross-domain imitation learning (Franzmeyer et al., 2022; Liu et al., 2023) offer promising avenues to address this challenge.

## 8 CONCLUSION

We introduce a new problem setting: few-shot IRL with multi-task data, which aims to learn from too few demonstrations in a task with variations by utilizing diverse multi-task demonstration data. We propose MPIRL, a novel method that tackles this problem by learning a two-part reward function: 1) a multi-task discriminator that uses the multi-task demonstrations to generalize over task variations and 2) a proximity reward that guides the policy in non-expert states. Finally, we demonstrate the effectiveness of our generalizable and well-shaped reward function in multiple navigation and manipulation environments, improving on the next best baseline by 33%.

REPRODUCIBILITY STATEMENT

To ensure our work is reproducible, we include our full codebase with example commands submitted as supplementary material with the data that we use available to download here. This codebase includes implementation of our method and all baselines, along with the demonstration datasets we used. In addition, we explain our method in detail in Section 4 and include additional implementation details about the environments and baselines in Appendix Section B and Appendix Section C.

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

# APPENDIX

## A  ADDITIONAL RESULTS

Figure 7 contain comparison results for additional FactorWorld tasks that did not fit into the main paper. Our method out-performs the baseline methods in every task and displays similar trends as those discussed in Section 6.1.

Figure 8 contains additional analysis experiments in different tasks for varying the number of target task demos and varying the number of tasks in the multi-task demo dataset. As discussed in further detail in Section 6.2, we see performance increasing with a moderate number of additional target demos. We also see generally that performance increases with more tasks in the multi-task demonstration dataset but seem to saturate at around 5-10 tasks for FactorWorld as additional tasks do not provide new information relevant to the target task.

## B  ENVIRONMENT DETAILS

### B.1  MAZE2D

We base our implementation on the Maze environment from the D4RL benchmark Fu et al. (2020). As show in Figure 3a, there are four balls placed in fixed locations, resulting in four tasks. The starting positions of the agent are randomly sampled. The state space is the agent's position, velocity, and positions of four balls, and then outputs an x- and y-velocity to navigate in the maze. Episodes have a horizon of 1500 timesteps. For the target task we use two demonstrations, and for the multi-task dataset we use 200 demonstrations for each of the remaining three tasks, all gathered by a planner-based policy provided in Pertsch et al. (2021).

### B.2  BLOCK STACKING

We use the implementation from Pertsch et al. (2021), there are five blocks on the ground with five different colors. The five block starting positions are randomly generated. In each task, the agent aims to pick up a block with color X and place it on a block with color Y (X and Y are two different colors selected from five colors). Different tasks have different pick-place colors. The state space contains the gripper's position, opening angle, velocity, and the position of the gripper fingers. It also includes the position and orientation of the block in quaternions. The action space consists of an (x, z)-displacement and a continuous action representing the degree of the robot gripper's opening. We collect 200 demonstrations for each task using a planner from Pertsch et al. (2021) and use 25 demonstrations for the target task. The target task is to stack the purple block on top of the blue block. The three tasks in the multi-task demonstration dataset are: purple on top of green, black on top of blue, and green on top of white. Episodes have a horizon of 500 timesteps.

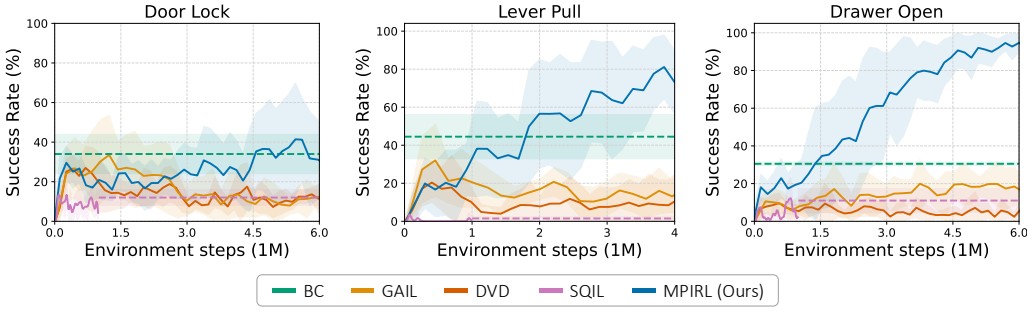

Figure 7: Remaining FactorWorld tasks that did not fit into the main paper. See Figure 4 for other tasks and experiment description.

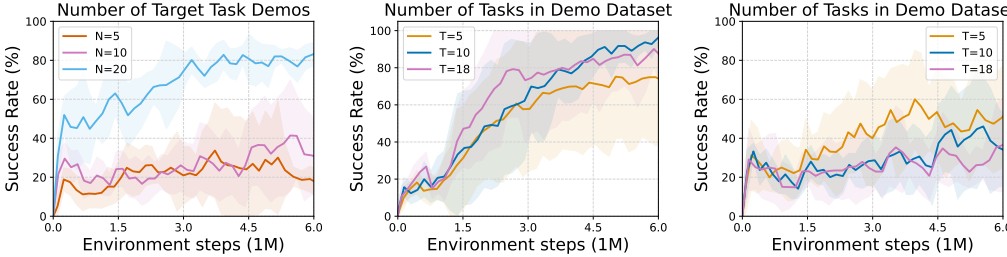

(a) Number of Target Task Demos  (b) Number of Tasks in $\mathcal{D}_{multitask}$  (c) Number of Tasks in $\mathcal{D}_{multitask}$

Figure 8: Additional analysis results over more tasks in supplement to Figure 5

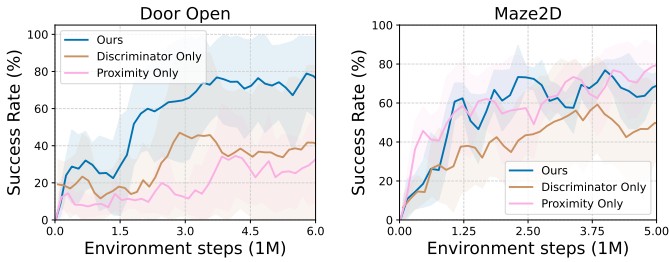

Figure 9: Ablations over more tasks in supplement to Figure 6a

### B.3 FACTORWORLD

We utilize the implementation provided by Xie et al. (2024), which extends the Meta-World benchmark (Yu et al., 2020) by introducing various factors of variations. In our experiments, we incorporate variations in object position, table position, and arm position, and include distractor objects with diverse initial positions and shapes. The agent observes in state space, the 3D position of its end effector, how open its gripper is, the 3D positions of the one or two objects on the tabletop, the goal position, and its previous state. The action space is the end effector position delta along with the normalized torque input to the gripper. We evaluate performance on seven tasks from the benchmark, using between 2 and 25 demonstrations for each task (Table 1). Since these tasks vary by difficulty, what is considered too few demonstrations varies. Additionally, we leverage an offline dataset consisting of 10 tasks randomly selected from the following set of 18 tasks, none of which are target tasks: reach, push, pick-place, dial-turn, drawer-close, button-press, peg-insert-side, window-open, sweep-into, basketball, door-close, faucet-open, hammer, handle-press-side, pick-out-of-hole, plate-slide, plate-slide-side, handle-pull. Each of these tasks has 200 demonstrations, collected by Meta-World's open-source hard-coded policies. The maximum number of timesteps per episode is capped at 500.

Table 1: FactorWorld Number of Target Demos

| Task | Drawer Open | Door Lock | Door Unlock | Plate Slide Back | Door Open | Lever Pull | Button Press Wall |
|---|---|---|---|---|---|---|---|
| # Demos | 5 | 10 | 5 | 5 | 2 | 25 | 10 |

## C IMPLEMENTATION DETAILS

We use the robot learning code base from `https://github.com/youngwoon/robot-learning` for basic RL and imitation learning baselines and use default hyperparameters unless otherwise specified. We detail our own implementations below.

## C.1   SQIL

We implement SQIL using the resources from Reddy et al. (2020) and use SAC (Haarnoja et al., 2018) as the off-policy RL algorithm. To incorporate the other task data, we add it to the training data with labeled rewards of 0. For each batch of training data, we sample 50% from target task demonstrations, 40% from the policy replay buffer, and 10% from the multi-task demonstrations. This addition can provide better coverage of the environmnet especially early on in training.

In environments where we used PPO (on-policy RL algorithm) for other IRL algorithms, we run SQIL until convergence, which often happened more quickly than the other methods because SAC tends to be more sample efficient than PPO. This is because SQIL requires an off-policy RL algorithm. While our method could also use SAC, in practice, we found the generative adversarial training for the multi-task discriminator to be more stable with PPO.

## C.2   DVD

We implement DVD and adapt the video-discriminator from the original paper to a state-action based reward function. Specifically, we input a demonstration trajectory including actions, and state-action tuple, and predict whether or not that state-action tuple exhibits expert behavior for the demonstrated task. Similar to our multi-task discriminator, we train DVD on $\mathcal{D} \cup \mathcal{D}_{multitask}$ using trajectory and state-action tuples from the same task as positive samples and trajectory and state-action tuples from different tasks as negative examples. We train DVD for 200 gradient steps using batch size of 128 and learning rate of 1e-3 then use it as a reward function to train a policy with online RL.

## C.3   MPIRL

We implement our method in two stages. First, we pretrain the actor in PPO policy, $\pi$, on $\mathcal{D}$ using a behavior cloning loss function 5 to mimic the behavior demonstrated in the dataset, providing a good initialization for subsequent policy training. Additionally, we pretrain the proximity reward $P$. Initially, we label the data from $\mathcal{D}$ as positive examples, and store data from $\mathcal{D}_{multitask}$ in the proximity dataset $\mathcal{D}_{prox}$, labeling them as negative examples. The pretraining is conducted over several epochs, with each epoch consisting of 50 iterations. At the end of each epoch, we perform a relabeling process: we randomly sample a batch of trajectories from the $\mathcal{D}_{prox}$ and select state-action tuples $(s_t, a_t)$ from each trajectory. These pairs are then relabeled based on the predictions from $P(s_t)$, which outputs continuous values in the range of [-1, 0]. For each trajectory, we apply backward relabeling from step $t$, assigning labels to earlier states as $prox(s_{t-k}) = P(s_t) - \gamma k$.

In the second stage, we fine-tune the policy adversarially in an online manner. Initially, we collect 2000 steps of policy data from the environment, storing this data in two separate buffers: policy data with predicted rewards $\tilde{R}(s, a)$ is stored in the policy replay buffer $\mathcal{D}_\pi$, and policy data with predicted proximity rewards $P(s)$ to the $\mathcal{D}_{prox}$. Once the data is stored, we begin training the discriminator $D(s, a)$ using Equation 1, the proximity reward $P(s)$ using Equation 2, and the policy $\pi$. This is followed by a relabeling process, similar to the one in the pretraining stage, with the exception if $D(s_t, a_t) > c_{thresh}$, those pairs are treated as expert states (positive examples) and excluded them from future relabeling. The second stage is repeated iteratively until the policy converges to the desired performance.

$$L_{BC} = \mathbb{E}_{(s,a)\sim\mathcal{D}} \|a - \pi(s)\|^2 \tag{5}$$

## C.4   HYPERPARAMETERS

For all environments we use a learning rate of 3e-4 for SAC and 1e-3 for the reward function. We use PPO with a clip ratio of 0.2 and batch size of 128. The proximity function is a feedforward network with 2 hidden layers of dimension 256 and tanh activation. The multi-task discriminator has the same architecture with an added lstm (2 layers, hidden dimension 128) to encode the demonstration trajectory, which is concatenated with the state-action tuple. The RL policy and critic are feedforward networks with 2 hidden layers of dimension 256 and relu activation.

Table 2: MPIRL hyperparameters.

| Hyperparameter | Maze2D | Block Stacking | FactorWorld |
|---|---|---|---|
| Proximity Reward Scale $\lambda_{prox}$ | 6 | 6 | 6 |
| $D$ threshold $c_{thresh}$ | 0.9 | 0.9 | 0.8 |
| Number of pretraining epochs | 5 | 100 | 5 |
| Proximity discount $\gamma$ | 0.001 | 0.001 | 0.001 |

