# OpenReview forum: "Learning Generalizable and Well-Shaped Reward Functions from Too Few Demonstrations"
_ICLR.cc/2025/Conference — ICLR 2025 Conference Withdrawn Submission_

### Official Review · Reviewer_EicC · 2024-10-16

**Soundness:** 2
**Presentation:** 2
**Contribution:** 2
**Rating:** 3
**Confidence:** 4

**Summary:**

This paper proposes a novel Inverse Reinforcement Learning (IRL) method called Multitask Discriminator Proximity-guided IRL (MPIRL). The method addresses the challenge of learning reward functions and reinforcement learning policies from too few demonstrations, which cannot fully specify the task in environments with variations. MPIRL leverages a multi-task dataset of expert trajectories to learn a generalizable and well-shaped reward function. Key components include a multi-task discriminator to infer expert behavior across task variations and a proximity reward that guides the agent toward the expert state distribution. This enables MPIRL to learn effective policies from limited demonstrations, outperforming previous few-shot IRL methods.

**Strengths:**

1. It is able to learn effective policies from limited demonstrations, outperforming previous few-shot IRL methods.
2. The idea of introducing the multi-task discriminator and proximity reward which gives the agent a well-shaped learning signal in non-expert states, guiding it back to the expert state distribution is interesting and sound.
3. This method can train a policy and learn a reward function simultaneously which can be beneficial to online tuning.

**Weaknesses:**

1. Lack of theoretical basis: you mention that you give a shaped reward but there is no theoretical analysis that can prove this reward can give the same optimal policy as the truth reward. CSIL[1] and OLLIE[2] both give a shaped reward from demonstrations that are guaranteed to provide the same optimal policy as the truth reward.
2. Lack of intuition: it is still unclear why the loss of discriminator is designed as Eqn (1). Why the former is for Task-specific Adversarial Training and the latter is for Multi-task Training?
3. Baselines out of time: The baselines are too old and too weak (the latest one in this paper is in 2021).
4. Lack of ablation study: you mention that this reward can be used for online RL. What will the result be like if you use an online RL method with this shaped reward?
5. Clarify of notations: D is used for the discriminator and the buffer which is confusing and hard to follow.
6. How do you implement \texttt{BC}, by minimizing the MSE loss in Eqn (5) or maximizing the likelihood?

[1] Watson, J., Sandy H., H., and Nicholas, H. Coherent soft imitation learning. In The 37th Conference on Neural Information Processing Systems, 2023.

[2] Yue, S., Hua, X., Ren, J., Lin, S., Zhang, J., and Zhang, Y. OLLIE: Imitation Learning from Offline Pretraining to Online Finetuning. The 41st International Conference on Machine Learning.

**Questions:**

1. Can this method be extended to utilizing data with diverse qualities as expert data is precious and hard to get?

See the Weaknesses.

---

### Official Review · Reviewer_oryX · 2024-11-02

**Soundness:** 3
**Presentation:** 2
**Contribution:** 2
**Rating:** 3
**Confidence:** 3

**Summary:**

In many imitation learning or inverse reinforcement learning settings, providing sufficient demonstrations of every scenario is one of the most tricky part in the overall pipelines. To address these difficulties, the authors propose Multitask Discriminator Proximity-guided IRL (MPIRL)

The proposed method, MPIRL utilize multi-task demo with few target task demo. MPIRL utilize two different reward term, proximity reward and discriminator reward. Proximity utilizes the expert for the reward signal, which could provide more rich signal to guide compared to using reward learners. Proximity reward P(s) penalizes states based on the temporal distance to the expert state distribution. Also, for the other part, MPIRL learns a adversarial discriminator for the multi-task demonstration. To learn the adversarial distriminator, MPIRL use state-action tuples from the same task as positive pairs, and tuples from different tasks or from the policy as negatives.

**Strengths:**

- In the three simulation environments, the proposed method improves over the baselines.

- Ablations and analysis show that using only proximity or discriminator terms are not enough to solve the problems. After combining them with proper scaling factor, MPIRL get the capability to solve the tasks with better shaped reward maps.

**Weaknesses:**

- The authors explains that they train a "generalizable and well-shaped reward function," but in the end, it seems that MPIRL simply extended the generative adversarial backbone used in GAIL to a multi-task setting. Also, there is no theoretical explanation in the paper to support the "well-shaped" aspect of the reward function. Even from looking at Figure 6, the Discriminator Reward or Proximity Reward doesn’t appear particularly well-shaped.


- There’s no sufficient explanation regarding finding the balance between the Discriminator reward and Proximity reward.

- The baselines all seem outdated. The most recent algorithm is DVD (Chen et al., 2021).

- There’s no ablation study about scaling factor lambda_prox.

**Questions:**

- Is there any theoretical grounding that supports the term “well-shaped”?

- How difficult it is to tune lambda_prox? Does the overall performance robust to lambda_prox? Is there any better way to find optimal lambda_prox value rather than just doing grid search?

- Is there any other recent baseline after DVD (Chen et al., 2021)?

- Could you please visualize the MPIRL reward maps, along with Discriminator only and Proximity only in Figure 6?

- Why MPIRL minus gamma for backwards re-labeling, rather than multiplying some value (e.g. 0.99).

- What exactly is the optimal policy, defined by the reward function that combines the Discriminator and Proximity functions, ultimately optimizing?

---

### Official Review · Reviewer_m4X6 · 2024-11-03

**Soundness:** 2
**Presentation:** 3
**Contribution:** 2
**Rating:** 3
**Confidence:** 3

**Summary:**

In this paper, the authors present an Inverse Reinforcement Learning (IRL) approach that learns policies from a multi-task demonstration dataset. Their method, named *Multitask Discriminator Proximity-guided IRL (MPIRL)*, uses a reward that combines a discriminator-based reward with a proximity-based reward. The proposed technique is evaluated on simulated navigation and manipulation tasks, demonstrating its performance and effectiveness compared to established baselines.

**Strengths:**

The paper addresses a relevant and important problem setup. The writing is mostly clear and easy to follow, facilitating a good understanding of the proposed approach. Additionally, the authors have made their code available, which enhances the reproducibility of their work.

**Weaknesses:**

The proposed method extends adversarial imitation learning by using a discriminator-based reward. However, the paper provides limited discussion on existing adversarial imitation learning approaches, aside from GAIL[1]. A more comprehensive overview and comparison of relevant adversarial imitation methods[2], along with a discussion on how different design choices influence learning, would strengthen the contextualization and emphasize the novelty of the work.

The discriminator $D(s, a)$ is trained on a multi-task dataset but relies on the current policy, making the reward signal relative rather than absolute. Consequently, the learned reward function therefore would not independently specify the task.. Addressing or clarifying this limitation would be beneficial.

Additionally, the term "too few demonstrations" should be more thoroughly explained in the introduction to provide readers with a clearer understanding of the experimental setup and motivation.

There is an issue in Equation 3: the inequality $> c_{thresh}$ is missing in the first condition. Ensuring that all equations are precise and complete is crucial for the technical clarity of the paper.

[1]Ho, J., & Ermon, S. (2016). Generative Adversarial Imitation Learning. In Advances in Neural Information Processing Systems
[2]Orsini, M., Raichuk, A., Hussenot, L., Vincent, D., Dadashi, R., Girgin, S., Geist, M., Bachem, O., Pietquin, O., & Andrychowicz, M. (2021). What Matters for Adversarial Imitation Learning? In Advances in Neural Information Processing Systems

**Questions:**

**Clarification on Value Calculation**: In lines 210-211, the authors state that $P(s)$ has a minimum value of -1. Could the authors elaborate on how this minimum value was derived or calculated? Providing a more detailed explanation or derivation would enhance the reader's understanding.

**Parameter Justification**: In Appendix C.4, it is mentioned that $\lambda_{prox} = 6$ for all tasks and that the proximity discount $\gamma = 0.001$. How were these specific values chosen, and what was the rationale behind setting these parameters? Additionally, could the authors discuss what the potential impact might be if $\gamma$ were set to a larger value? Exploring this would help clarify the sensitivity and robustness of the proposed approach to these hyperparameters.

**Consideration of Alternative Algorithms**: Have the authors considered using other reinforcement learning algorithms, such as Soft Actor-Critic (SAC), to optimize the proposed reward function in Equation 4? An analysis or discussion on the choice of RL algorithm and the potential benefits or drawbacks of using alternatives like SAC would be insightful.

**Clarification on Few-shot IRL vs. Meta-learning**: In Section 2.2, the authors discuss the distinction between Few-shot IRL and meta-learning. However, this difference is not entirely clear. Could the authors provide a more in-depth explanation or additional context to better articulate how these two concepts differ?

---

### Official Review · Reviewer_FyAf · 2024-11-04

**Soundness:** 3
**Presentation:** 3
**Contribution:** 2
**Rating:** 6
**Confidence:** 2

**Summary:**

This paper proposes Multitask Discriminator Proximity-guided IRL (MPIRL), an approach that uses an multi-task demo dataset to address the few-shot inverse RL problem when the agent only has access to a few task-specific demonstrations which may not fully specify the task to solve. To guide the policy back to the expert state distribution when in non-expert states, the authors propose a temporal distance-based proximity reward term that is continually updated over policy training.

**Strengths:**

The paper is generally well-written. The proposed idea is intuitive and well-explained. The experiments demonstrate substantial improvement over other inverse RL and imitation learning baselines.

**Weaknesses:**

- The paper would benefit from additional baselines, such as a meta-learning baseline as discussed in the related work section. It would also be nice to include an expert / oracle baseline for comparison, such as PPO trained on the ground-truth reward function.

- It would also be interesting to evaluate the scalability of the proposed approach on more challenging multi-task domains, such as RLBench (https://arxiv.org/abs/1909.12271)

- How necessary is staged training of MPIRL (Appendix C.3)? How well does your method work without staged training? Furthermore, these details should be mentioned at minimum in the main body of the paper, and they are not included in Algorithm 1 currently.

**Questions:**

- How robust is MPIRL to the choice of $\lambda_{prox}$ and $c_{thresh}$? Some ablations comparing different choice of values for these hyperparameters would be helpful.

- How well does MPIRL work with $N=1$ target task demos?

---

### Note · Authors · 2024-11-15

**Comment:**

We thank all the reviewers for their thorough and helpful reviews.  Given the general agreement between reviewers we will withdraw and work on incorporating their feedback to improve our paper for a future submission.

**Withdrawal Confirmation:**

I have read and agree with the venue's withdrawal policy on behalf of myself and my co-authors.